# Establishment and Calibration of a Digital Twin to Replicate the Friction Behaviour of a Pin-on-Disk Tribometer

Erik Hansen [1,*] , Gerda Vaitkunaite [2] , Johannes Schneider [2] , Peter Gumbsch [2] and Bettina Frohnapfel [1]

1   Institute of Fluid Mechanics (ISTM), Karlsruhe Institute of Technology (KIT), 76131 Karlsruhe, Germany
2   Institute for Applied Materials—Reliability and Microstructure (IAM-ZM), Karlsruhe Institute of Technology (KIT), 76131 Karlsruhe, Germany
*   Correspondence: erik.hansen@kit.edu

**Abstract:** While the modification of surface contacts offers significant potential for friction reduction, obtaining an underlying consistent friction behaviour of real-life experiments and virtual simulations is still an ongoing challenge. In particular, most works in the literature only consider idealised geometries that can be parametrised with simple analytical functions. In contrast to this approach, the current work describes the establishment of a digital twin of a pin-on-disk tribometer whose virtual geometry is completely replicated from real-life post-test topography measurements and fed into a two-scale mixed lubrication solver. Subsequently, several calibration steps are performed to identify the sensitivities of the friction behaviour towards certain geometry features and enable the digital twin to robustly represent the Stribeck curve of the physical experiments. Furthermore, a derivation of the Hersey number is used to generalise the obtained friction behaviour for different dynamic viscosities and allow the validation of the presented method.

**Keywords:** digital twin; pin-on-disk tribometer; mixed lubrication; multi-scale modelling

## 1. Introduction

The field of tribology offers a vast amount of approaches to reduce friction losses in many applications by altering surface contacts [1]. Exemplary approaches are the adding of surface coatings [2], tribofilms [3,4] or surface textures [5]. Due to the high sensitivity of these approaches, generally applicable design guidelines are difficult to obtain. For example, in the case of surface textures, the design usually needs to be tailored to the specific conditions within the tribological contact to exploit their full potential [5,6]. These conditions can be most clearly defined and controlled in tribometers, where different kinds of tribometers represent specific tribological contacts that occur in certain applications. Ball-on-disk tribometers are, for example, used to represent the non-conformal contacts of ball bearings while pin-on-disk tribometers are used to mimic the conformal contacts of journal bearings. The similarity of the tribological contact eventually allows to transfer the fundamental insights obtained in tribometers to applications.

Due to the highly controlled operating conditions and reduction to the tribological contact, tribometers are easier to simulate with a numerical model than their corresponding applications. Nonetheless, even under these simplified circumstances, the modelling of the tribological contact is not straightforward and consistent results from simulations and experiments are challenging to obtain [7]. Despite these difficulties, the potential of a joint experimental and numerical research strategy is significant because it allows to obtain complementary in and ex situ data. Furthermore, the simulations can be used to perform cheap and quick a priori investigations to identify the relevant parameter ranges for the experiment. The obtained simulation results can then be confirmed by selectively chosen experiments [8]. This allows to conduct targeted, efficient investigations and to save costs due to material consumption and the associated transportation. Moreover, the experiment results give validation and new impulses for the adjustment of the simulation model and

parameters. This process can be further improved by employing meta-models to select the simulation parameters such that the prediction quality of the investigation is maximised. This in turn allows to reduce the necessary amount of simulations and allows more complex models to be used [9].

The above described interaction between the experiment and simulation generally corresponds to the concept of a digital twin [10]. While some works strictly require a digital twin to have bi-directional real-time communication between the physical and virtual entity [11], other works already consider a one-directional physical-to-virtual connection as sufficient to be labelled as a digital twin [12]. The latter case also applies to this work, in which a digital twin of a pin-on-disk tribometer is established and calibrated in several steps. Firstly, the physical entity of the pin-on-disk tribometer is described along with the associated sample preparation, testing procedure and topography measurements. Subsequently, the macro- and microscopic reconstruction of the virtual geometry from the real-life post-test topographies is presented and the employed two-scale mixed lubrication solver is described. Afterwards, the experimental results are evaluated and virtual calibration steps are performed to obtain a representative virtual geometry for an arbitrary specimen pairing. At the same time, the calibration results are used to identify the macro- and microscopic sensitivities of the digital twin. Moreover, the Hersey number is derived for a pin-on-disk tribometer to allow for the validation of the digital twin with the Stribeck curves of its real-life counterpart and the generalisation to operating conditions at different dynamic viscosities. Exemplary codes and geometry files are included in the supplementary material to enable the public availability of the exact implementation of the digital twin. The purpose of this work is firstly to demonstrate the achievable agreement of a real-life experiment and its digital twin and secondly to provide a detailed example as orientation for anyone who wants to create a digital twin of tribological contacts in mixed lubrication.

## 2. Methods

### 2.1. Physical Entity of the Pin-on-Disk Tribometer

2.1.1. Sample Preparation and Tribometer Tests

The experiments were carried out in a pin-on-disk tribometer (Plint TE-92 HS, Phoenix Tribology, Kingsclere, UK). The pin and disk samples were made from 100Cr6 bearing steel. Before each measurement, the samples were cleaned for 5 min in an ultra-sonic bath with isopropanol to remove manufacturing residuals and debris.

The pin diameter was 8 mm and the disk diameter was 80 mm. Pre-test pin and disk roughness were measured with a contact measurement device (Hommel T8000, Jenoptic, Jena, Germany) with a 2 μm conical-shaped diamond stylus. Roughness values ($Ra$) were acquired by tracking a 7 mm profile across the samples. Three measurements are taken per sample to ensure the repeatability of the initial sample surface. The averaged sample roughness before the tests was $Ra = 0.2$ μm and $Ra = 0.07$ μm for the pin and disk samples, respectively. The hardness of the disk was 800 HV and the pin hardness is 713 HV.

As shown in Figure 1a, the pin sample was fixed into the half-sphere holder, thus protruding out of the pin holder by a pin height of 1 mm for all experiments. The disk sample was mounted on the tribometer platform, which was connected to the motor. Before each test, the disk surface waviness was evaluated by contact stylus to avoid mounting errors and a subsequent potential influence on the acquired friction values. The waviness of the disk did not exceed 1 μm after being mounting onto the tribometer platform.

Additionally, to avoid mounting errors of the pin, the self-alignment mechanism of the pin was used to level the pin. The schematics of the pin levelling process are given in Figure 1a. Firstly, the samples in the tribometer were pre-loaded under an initial imposed normal load of $F_{N,imp} = 50$ N, then 5 mL of oil were injected beneath the pin holder. The force generated under the pin holder pushed the pin towards the disk surface and the pin aligned itself against the disk. Once the pressurised oil flow underneath the pin stopped, the pin holder settled again in its base, thus fixing the obtained alignment for the remainder of the experiment.

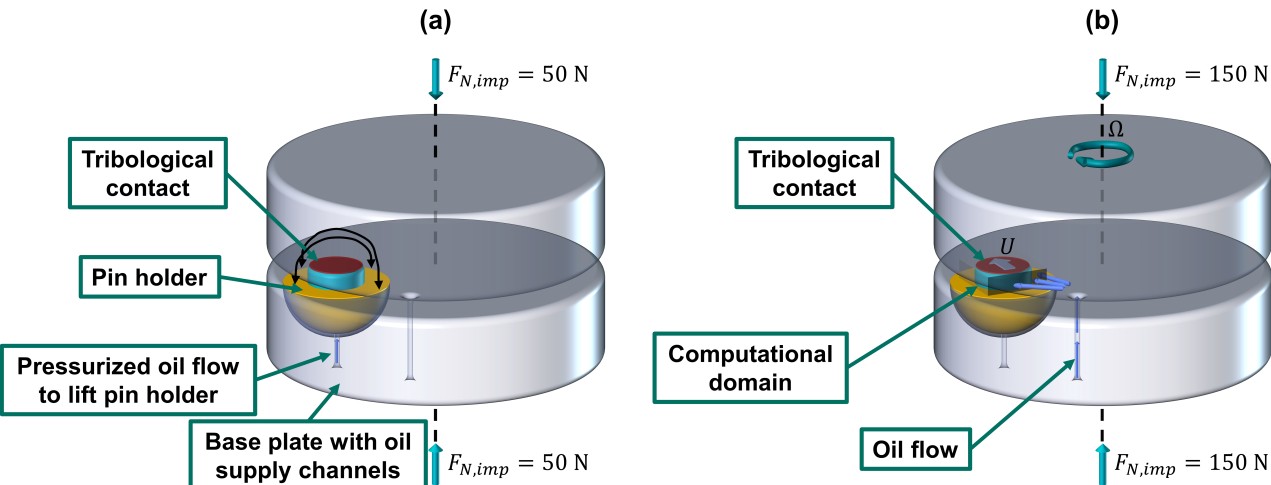

**Figure 1.** Schematic depiction of (**a**) pin levelling process and (**b**) Stribeck curve acquisition in the pin-on-disk tribometer.

The friction behaviour was evaluated by acquiring the Stribeck curve as depicted in Figure 1b at an imposed normal load of $F_{N,imp} = 150$ N. The tribometer tests started at 2 m/s and the sliding speed gradually decreased to 0.04 m/s. The selected sliding speed range enables the analysis of the lubrication regime shift from hydrodynamic to mixed lubrication. Each sliding speed step is held for 5 min to record the friction data. The speed ramps are repeated five times for each experiment. To determine the final Stribeck curve of the whole test, only the last three ramps were averaged to avoid running-in effects at the beginning of the experiment.

An additive-free mineral base oil was chosen to investigate the hydrodynamic/mixed lubrication transition. The detailed testing matrix is given in Table 1 and the oil properties are presented in Table 2. The dynamic viscosity is measured with a Discovery Hybrid Rheometer (DHR series, TA instruments, New Castle, USA) at temperatures between 24 and 50 °C and at a shearing rate of 500 s$^{-1}$. The final dynamic viscosity value at each temperature was the average of 50 measurements.

The experiments were conducted at lubricant temperatures of 24 and 50 °C to investigate the viscosity effect. In combination with the imposed normal load, these operating conditions only led to mild wear effects that did not exceed the simple running in of the pin and disc. Higher loads and temperatures, on the other hand, would cause significant wear scars, which in turn largely alter the contact geometry and would thus not deliver comparable data. The oil flow in the system was kept constant at 5 mL/min for all experiments. To ensure the repeatability of the friction behaviour, each experiment was repeated 3 times with new samples and fresh oil in the system. Thus, two experiment sets with three tests each were conducted. The design of experiments is given in Table 3 with the corresponding dynamic viscosities.

**Table 1.** Tribometer testing parameters.

| | Constant Parameters | |
| :---: | :---: | :---: |
| **Normal Load (N)** | **Sliding Radius (mm)** | **Oil Flow Rate (mL/min)** |
| 150 | 30 | 5 |
| | Changing Parameters | |
| **Speed (m/s)** | **Oil Temperature (°C)** | |
| 2 . . . 0.04 | 24, 50 | |

**Table 2.** Properties of the oil.

| Properties | Value |
|---|---|
| Kinematic viscosity (mm$^2$/s) at 40 °C [1] | 29 |
| Kinematic viscosity (mm$^2$/s) at 50 °C [1] | 20 |
| Dynamic viscosity (Pas) at 24 °C [2] | 0.066 |
| Dynamic viscosity (Pas) at 50 °C [2] | 0.024 |
| Viscosity index (ISO 2909) [1] | 101 |
| Density (g/ml) at 15 °C [1] | 0.87 |

[1]—measurements provided by lubricant manufacturer. [2]—measurements conducted at Karlsruhe Institute of Technology (Karlsruhe, Germany).

**Table 3.** Design of experiments.

| Set | Test | Temperature (°C) | Dynamic Viscosity (Pas) |
|---|---|---|---|
| 1 | 1.1 1.2 1.3 | 24 | 0.066 |
| 2 | 2.1 2.2 2.3 | 50 | 0.024 |

2.1.2. Post-Analysis of the Surfaces

After the tribometer tests, the samples were cleaned again for 5 min in an ultra-sonic bath with isopropanol. Subsequently, the macroscopic sample topographies of the pin and disk were measured with the confocal microscope FRT (Fries Research & Technology GmbH, Bergisch Gladbach, Germany). The surface topography measurement size was 8 mm × 8 mm with a 500 × 500 pixel resolution per single measurement. The selected measurement area allowed the acquisition of the whole surface of the tested pin and the corresponding contact area on the worn disk. Furthermore, microscopic roughness topography patches of pin and disc were obtained with the white light interferometry profiler Sensofar (Sensofar Metrology, Barcelona, Spain). The measurement resolutions for the pin and disk samples were 768 × 576 pixel for an analysis area of 84.9 µm × 63.7 µm.

*2.2. Digital Twin of the Pin-on-Disk Tribometer*

The establishment of the digital twin of the pin-on-disk tribometer consisted of two parts which are elaborated upon in the following subsections. The first part describes how the virtual geometry was reconstructed in the macroscopic and microscopic roughness scales from the post-test topographies of the real-life pin and disk. The second part highlights the fundamentals of the solver that was used to simulate the Stribeck curve acquisition.

2.2.1. Virtual Geometry Reconstruction

The macro- and microscopic post-test topographies were first processed with the software Gwyddion$^©$ to correct the measurement defects through interpolation with the Laplace equation, rotate the topography towards its mean plane as an initial levelling guess, set the zero-height mark to the mean plane and limit the height range to 20 µm. In case of the macroscopic pin topography, the points belonging to the area surrounding the pin were excluded in the computation of the mean plane. Furthermore, a Gaussian filter was applied to both the macroscopic pin and disk topographies to remove roughness information and measurement noise by smoothing the profiles. The size of the Gaussian filter was parametrised by its full width at half maximum of the Gaussian distribution. The microscopic roughness topographies, on the other hand, did not require any filtering

since their measured profiles were sufficiently smooth due to the higher resolution. The remaining processing was performed with MATLAB$^{©}$. On both the macro- and microscopic scale, the disk topography was flipped above the pin and the topographies were aligned such that the $x_1$ axis pointed in the same direction as the velocity of the disk $U$. In the case of the macroscopic topographies, the origin of the coordinate system was also aligned with the centre of the pin. For the macroscopic pin, the pin height of 1 mm was set relative to its zero-height mark. Since any profile variations in the disc and the pin holder were assumed to be negligible in comparison to the overall pin height, the profiles of the disc and the pin holder around the pin were set to be perfectly smooth. Lastly, the resolution of the macroscopic profiles was adjusted by interpolation. For the roughness topographies, a subdomain of 256 × 256 pixel was extracted without interpolation from the original topography. The effect of the filtering on the resulting macroscopic virtual geometry of test 2.1 is depicted in Figure 2 for an exemplary rigid body displacement of 10 µm and resolutions of 500 × 500 pixel, where unfiltered topographies are used in (a) and a 9 pixel Gaussian filter is used in (b).

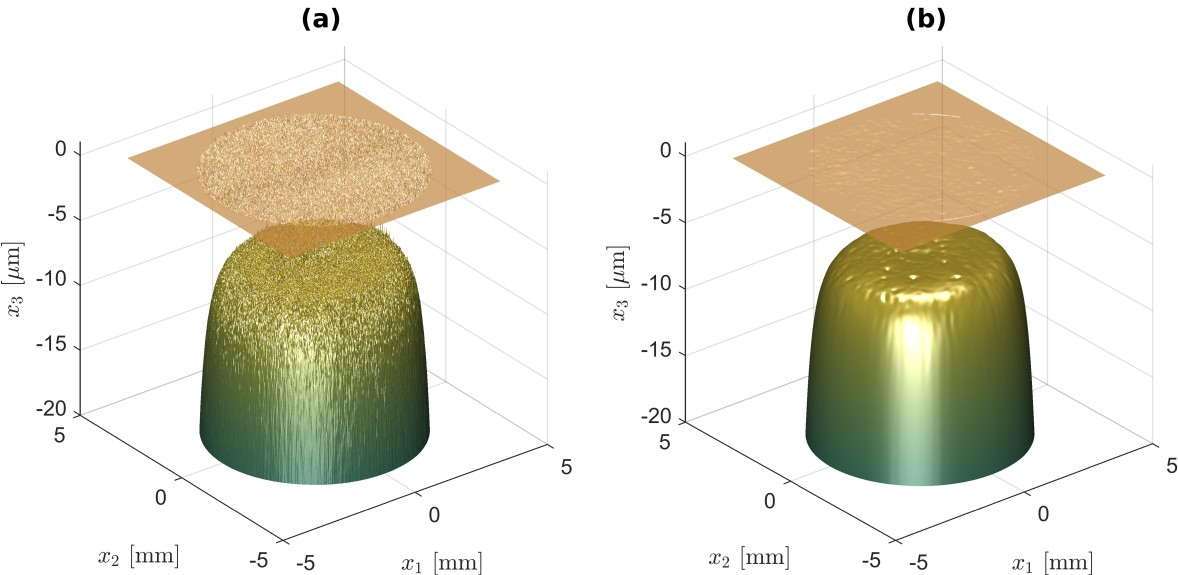

**Figure 2.** Exemplary virtual macroscopic geometry of test 2.1, where the disk is represented by the top surface and the pin by the bottom surface. Note the different scale of the vertical and the horizontal axes. Below the shown $x_3$ range, the pin holder profile is modelled as a flat surface at $x_3 = -1$ mm: (**a**) without filtering; and (**b**) after applying a 9 pixel Gaussian filter.

In order to simulate the alignment of the pin against the disk at an imposed normal load of $F_{N,imp} = 50$ N as shown in Figure 1a, a dry contact solver for non-periodic problems was coupled with the torque balance equation. The dry contact solver is based on the conjugate gradient-fast Fourier transform (CG-FFT) algorithm described by Polonsky and Keer [13] and Sainsot and Lubrecht [14]. During inner iterations, the code computes the equilibrium of elastic deformation and dry contact pressure for an imposed normal load while using linear convolutions with the kernel function derived from the elastic half-space theory when constant pressure over rectangular discretisation cells is assumed [15] (Ch. 3.3), [16]. Young's modulus $E$ and the Poisson ratio $\nu$ of the upper and lower surfaces are considered in the kernel function. Within the contact region, the dry contact pressure can take values between zero as a lower limit and the hardness $H$ of the material as a maximum limit [17]. Outside of the contact zone, the pressure is set to zero. Once the dry contact solver delivers a converged dry contact pressure field $p_{con,dry}(x_1, x_2)$, this field is

used in outer iterations to evaluate the resulting torques $T_{x_1}$ and $T_{x_2}$ around the $x_1$ and $x_2$ axes by summing up over all of the $N_{x_1}N_{x_2}$ grid points of size $\Delta x_1 \Delta x_2$:

$$T_{x_1} = \sum_{N_{x_1}} \sum_{N_{x_2}} -x_2 p_{con,dry} \Delta x_1 \Delta x_2, \tag{1}$$

$$T_{x_2} = \sum_{N_{x_1}} \sum_{N_{x_2}} x_1 p_{con,dry} \Delta x_1 \Delta x_2. \tag{2}$$

The dimensionless residuals of the torque balances are computed at each outer iteration $n$ as:

$$r_{x_1}^n = \frac{T_{x_1}^n}{\sum_{N_{x_1}} \sum_{N_{x_2}} \mathrm{abs}\left((-x_2) p_{con,dry}^n \Delta x_1 \Delta x_2\right)}, \tag{3}$$

$$r_{x_2}^n = \frac{T_{x_2}^n}{\sum_{N_{x_1}} \sum_{N_{x_2}} \mathrm{abs}\left(x_1 p_{con,dry}^n \Delta x_1 \Delta x_2\right)}. \tag{4}$$

As long as these residuals are larger than a prescribed tolerance of $10^{-6}$, angles $\alpha_{x_1}$ and $\alpha_{x_2}$ are adjusted by a PID controller with its coefficients $K_P$, $K_I$ and $K_D$:

$$\alpha_{x_1} = \left( K_P r_{x_1}^n + K_I \sum_i^n r_{x_1}^i + K_D \left( r_{x_1}^n - r_{x_1}^{n-1} \right) \right) \cdot 360°, \tag{5}$$

$$\alpha_{x_2} = \left( K_P r_{x_2}^n + K_I \sum_i^n r_{x_2}^i + K_D \left( r_{x_2}^n - r_{x_2}^{n-1} \right) \right) \cdot 360°. \tag{6}$$

Afterwards, the new pin profile $x_{3,low}$ is computed by altering the unlevelled pin profile $x_{3,low,0}$ and the loop is repeated:

$$x_{3,low} = x_{3,low,0} + x_2 tan(\alpha_{x_1}) - x_1 tan(\alpha_{x_2}). \tag{7}$$

Note, however, that $x_{3,low} = -1$ mm is enforced for the area around the pin because there, the levelling is assumed to be negligible in comparison to the pin height. The described levelling process is performed for each of the virtual macroscopic gap geometries. The employed values of the solver parameters are summarised in Table 4.

**Table 4.** Employed parameter values of the pin levelling solver.

| Parameter | Value |
|:---:|:---:|
| $F_{N,imp}$ | 50 N |
| $\nu_{up}$ | 0.3 |
| $\nu_{low}$ | 0.3 |
| $E_{up}$ | $210 \cdot 10^9$ Pa |
| $E_{low}$ | $210 \cdot 10^9$ Pa |
| $H$ | $6.99 \cdot 10^9$ Pa |
| $K_P$ | $3 \cdot 10^{-6}$ |
| $K_I$ | $6 \cdot 10^{-6}$ |
| $K_D$ | $3.75 \cdot 10^{-7}$ |

Before the profiles are averaged, the mean plane of each levelled pin profile within a radius of 3 mm from its centre is used to define its new zero-height mark. This is important to reduce the effect of the pin rim on the mean plane which could otherwise distort the following averaging by over-weighting certain pin topographies. For the disks, the zero-height mark remains unchanged. Once this is done for all of the tests within a set, the geometries are interpolated on a new grid on which the set average is determined for pin

and disk. Below a truncation height of $-20$ µm, the pin height of $-1$ mm is enforced again. Finally, the obtained averaged geometry is levelled again. The centre lines of the resulting virtual macroscopic pin topographies of set 2 in the exemplary case of perfectly smooth macroscopic disk topographies as counter bodies are displayed in Figure 3.

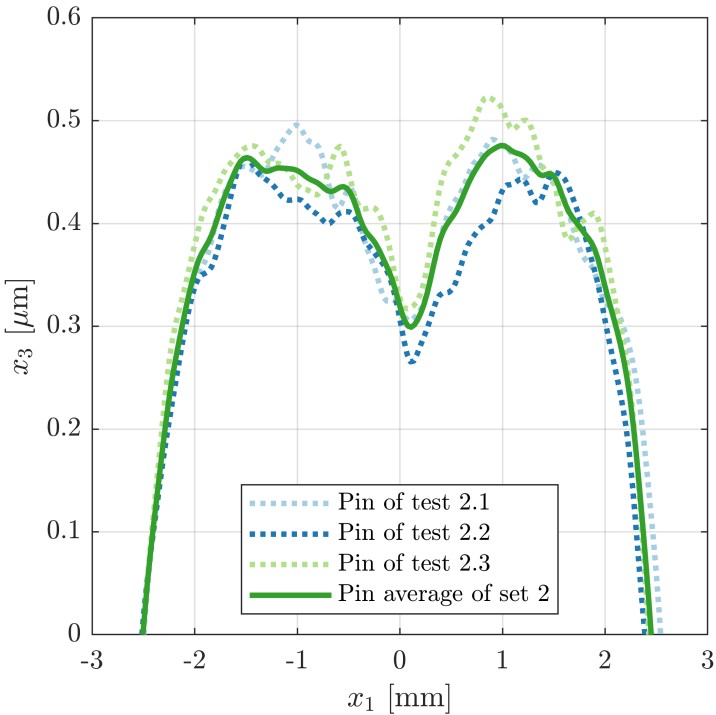

**Figure 3.** Center line plot of the virtual macroscopic pin topographies of tests 2.1, 2.2, 2.3 and the average of set 2. Note the different scale of the vertical axis. The topographies are previously filtered with a 9 pixel Gaussian filter and the resolution is adjusted to $256 \times 256$. The levelling is performed for perfectly smooth macroscopic disk counter bodies.

Similarly to some concepts in the literature [18–21], the microscopic geometries are used to compute the mean contact pressures, mean gap heights and homogenisation factors for 100 distinct rigid body displacements on the roughness scale $h_0$ in an exponentially spaced range between $h_{0,r,min} = 0.1$ µm and $h_{0,r,max} = 1$ µm. The averaging is performed over the periodic roughness domain lengths $L_{r,1}$ and $L_{r,2}$ and the periodic roughness time length $T_r$. For each rigid body displacement, $N_t = 32$ discrete time steps are used to periodically move the roughness profile of the disk in the $x_1$ direction once over the roughness profile of the pin. At each time $t$, the asperity contact pressure $p_{asp}$ and deformed gap height distributions $h$ are computed at each spatial position $(x_1, x_2)$ with a dry contact solver for periodic problems. It is of CG-FFT type and its algorithm is mainly based on the description by Akchurin et al. [22]. The code computes the equilibrium of elastic deformation and asperity contact pressure for the imposed $h_0$ while using cyclic convolutions with the kernel function derived from the elastic half-space theory when constant pressure over rectangular discretisation cells is assumed [15] (Ch. 3.3), [16]. The Young's modulus $E$ and Poisson ratio $\nu$ of the upper and lower surfaces are considered in the kernel function. Within the contact region, the asperity contact pressure can take values between zero as a lower limit and the hardness $H$ of the material as a maximum limit [17]. Outside of the contact zone, the pressure is set to zero. The employed values of Young's modulus, Poisson ratio and hardness are the same as those used for the levelling solver in Table 4.

The obtained gap height distribution $h$ is truncated below 1 nm and used to compute solutions $\chi_1$, $\chi_2$ and $\chi_3$ of the unsteady local problems:

$$\nabla \cdot \left( h^3 \nabla \chi_1 \right) = \frac{\partial h}{\partial x_1} + \frac{1}{u_m} \frac{\partial h}{\partial t}, \tag{8}$$

$$\nabla \cdot \left( h^3 \nabla \chi_2 \right) = -\frac{\partial h^3}{\partial x_1}, \tag{9}$$

$$\nabla \cdot \left( h^3 \nabla \chi_3 \right) = -\frac{\partial h^3}{\partial x_2}. \tag{10}$$

For the first time step, the steady problem is solved by neglecting the unsteady term. The local problems are discretised with the finite volume method (FVM). Second-order central schemes are used for the spatial derivatives, while the first-order Euler implicit scheme is used for the temporal derivative. Periodic boundary conditions are employed. One point of the domain is used for the Dirichlet condition $\chi_1 = \chi_2 = \chi_3 = 0$. The value of the Dirichlet condition can be chosen arbitrarily and does not influence the final homogenisation factors since they are only a function of the gradients of $\chi_1$, $\chi_2$ and $\chi_3$. Furthermore, the discretised form of Equation (8) becomes independent of $u_m$ by setting the time step size to $L_{r,1}/(N_t u_m)$, thus cancelling out $u_m$ in the final expression. Once all time steps are solved for, their average is computed to obtain the mean contact pressure $p_{con}$, mean gap height $h_m$ and the homogenisation factors $A$, $\vec{b}$, $C$ and $\vec{d}$:

$$p_{con}(h_0) = \frac{1}{L_{r,1} L_{r,2} T_r} \int\limits_{L_{r,1}} \int\limits_{L_{r,2}} \int\limits_{T_r} p_{asp} \, \mathrm{d}t \, \mathrm{d}x_2 \, \mathrm{d}x_1 \tag{11}$$

$$h_m(h_0) = \frac{1}{L_{r,1} L_{r,2} T_r} \int\limits_{L_{r,1}} \int\limits_{L_{r,2}} \int\limits_{T_r} h \, \mathrm{d}t \, \mathrm{d}x_2 \, \mathrm{d}x_1 \tag{12}$$

$$A(h_0) = \frac{1}{L_{r,1} L_{r,2} T_r} \int\limits_{L_{r,1}} \int\limits_{L_{r,2}} \int\limits_{T_r} \frac{h^3}{h_m^3} \begin{pmatrix} 1 + \frac{\partial \chi_2}{\partial x_1} & \frac{\partial \chi_3}{\partial x_1} \\ \frac{\partial \chi_2}{\partial x_2} & 1 + \frac{\partial \chi_3}{\partial x_2} \end{pmatrix} \mathrm{d}t \, \mathrm{d}x_2 \, \mathrm{d}x_1 \tag{13}$$

$$\vec{b}(h_0) = \frac{1}{L_{r,1} L_{r,2} T_r} \int\limits_{L_{r,1}} \int\limits_{L_{r,2}} \int\limits_{T_r} \frac{h}{h_m} \begin{pmatrix} 1 - h^2 \frac{\partial \chi_1}{\partial x_1} \\ -h^2 \frac{\partial \chi_1}{\partial x_2} \end{pmatrix} \mathrm{d}t \, \mathrm{d}x_2 \, \mathrm{d}x_1 \tag{14}$$

$$C(h_0) = \frac{1}{L_{r,1} L_{r,2} T_r} \int\limits_{L_{r,1}} \int\limits_{L_{r,2}} \int\limits_{T_r} \frac{h}{h_m} \begin{pmatrix} 1 + \frac{\partial \chi_2}{\partial x_1} & \frac{\partial \chi_3}{\partial x_1} \\ \frac{\partial \chi_2}{\partial x_2} & 1 + \frac{\partial \chi_3}{\partial x_2} \end{pmatrix} \mathrm{d}t \, \mathrm{d}x_2 \, \mathrm{d}x_1 \tag{15}$$

$$\vec{d}(h_0) = \frac{1}{L_{r,1} L_{r,2} T_r} \int\limits_{L_{r,1}} \int\limits_{L_{r,2}} \int\limits_{T_r} h \, h_m \begin{pmatrix} \frac{\partial \chi_1}{\partial x_1} \\ \frac{\partial \chi_1}{\partial x_2} \end{pmatrix} \mathrm{d}t \, \mathrm{d}x_2 \, \mathrm{d}x_1 \tag{16}$$

Once these factors are computed for several different roughness patches of a single post-test pin and disk specimen combination, the test average is computed. The results of six different pin and disk patches of test 2.1 and their average are displayed for the mean gap height $h_m$ in Figure 4a, the mean contact pressure $p_{con}$ in Figure 4b and the homogenisation factor $A_{11}$ in Figure 5a. The averages of the homogenisation factors $A$ and $\vec{b}$ over the six patches are depicted in Figure 5b. The root-mean-square values of the roughness topography of each patch are provided in Table 5. The reconstructed microscopic virtual geometry of patch 4 of test 2.1 is exemplarily shown in Figure 6 for a rigid body displacement of 1 μm.

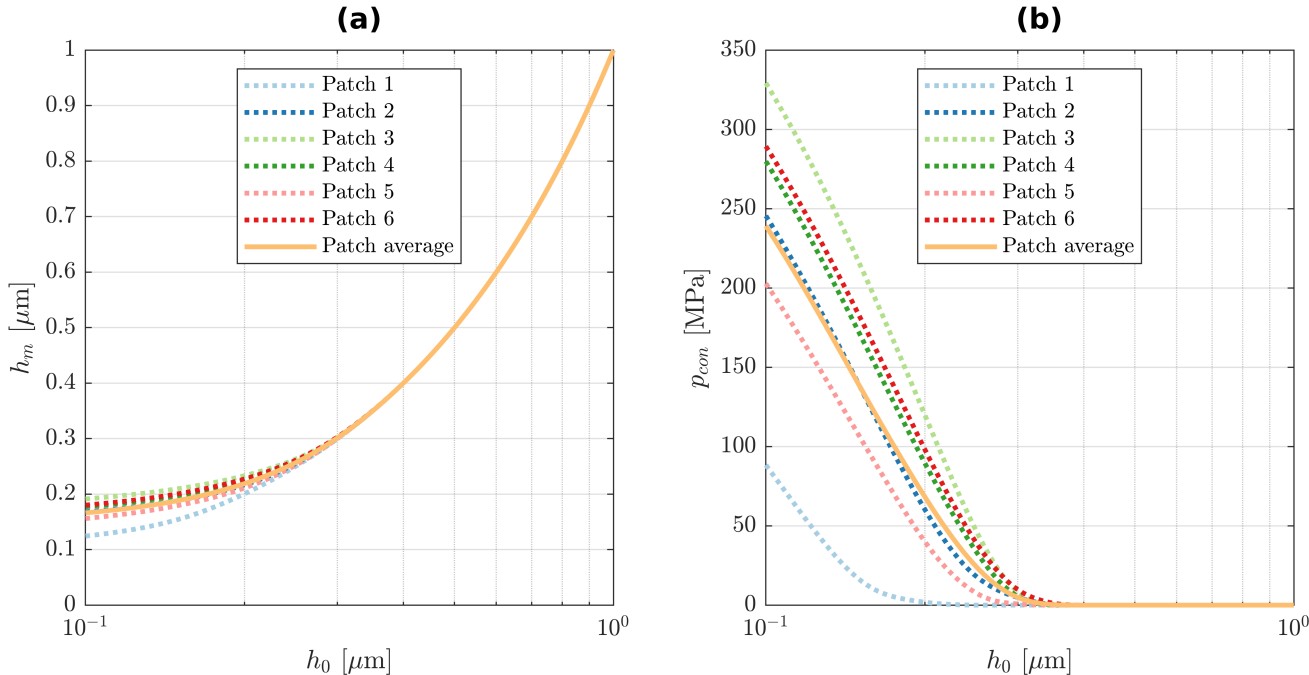

**Figure 4.** (**a**) Mean gap height $h_m$ and (**b**) mean contact pressure $p_{con}$ as a function of rigid body displacement $h_0$ for different roughness patches and their average.

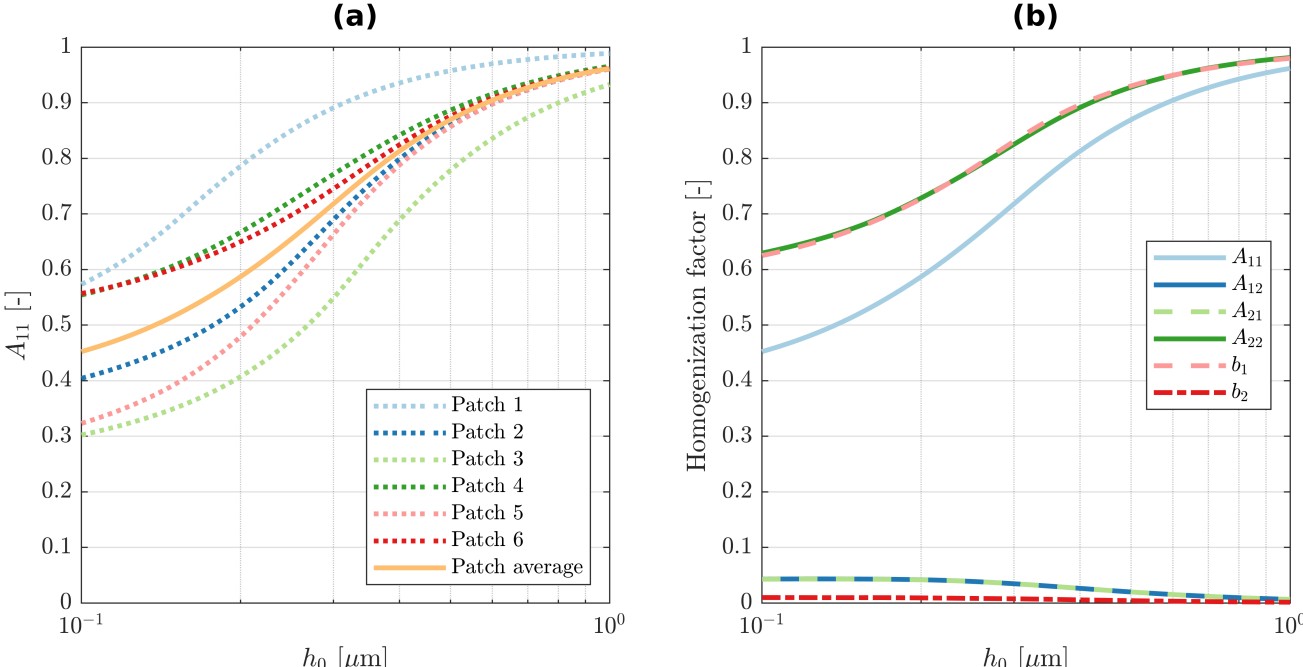

**Figure 5.** (**a**) Homogenisation factor $A_{11}$ as a function of rigid body displacement $h_0$ for different roughness patches of test 2.1 and their average. (**b**) Averages of the homogenisation factors $\boldsymbol{A}$ and $\vec{b}$ over test 2.1 as a function of rigid body displacement $h_0$.

**Table 5.** Root-mean-square values $S_q$ of the roughness patches of test 2.1, where the zero-height mark of each surface is at its respective mean plane.

| Patch | $S_{q,up}$ | $S_{q,low}$ | $S_q = \sqrt{S_{q,up}^2 + S_{q,low}^2}$ |
|---|---|---|---|
| 1 | 84.5 nm | 41.5 nm | 94.1 nm |
| 2 | 84.6 nm | 116.5 nm | 144.0 nm |
| 3 | 118.3 nm | 132.3 nm | 177.5 nm |
| 4 | 119.9 nm | 101.4 nm | 157.0 nm |
| 5 | 96.3 nm | 96.7 nm | 136.4 nm |
| 6 | 100.1 nm | 131.2 nm | 165.0 nm |

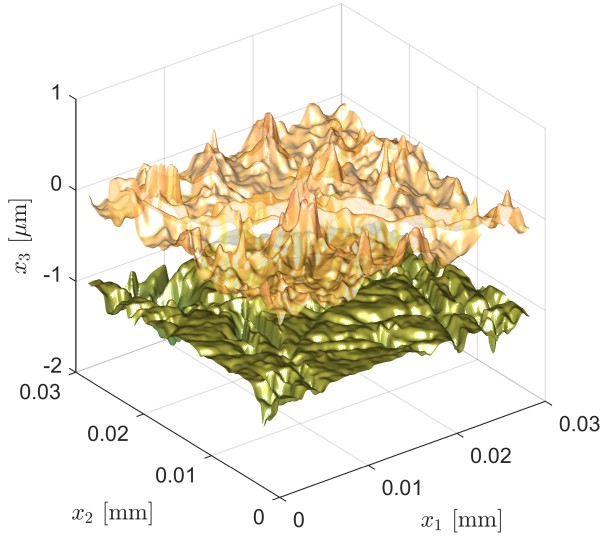

**Figure 6.** Exemplary virtual microscopic geometry of patch 4 of test 2.1, where the disk is represented by the top surface and the pin by the bottom surface. Note the different scale of the vertical and the horizontal axes.

### 2.2.2. Simulation of the Stribeck Curve Acquisition

In order to simulate the conditions within the macroscopic scale of the lubrication flow during the Stribeck curve acquisition depicted in Figure 1b, a homogenised mixed elasto-hydrodynamic lubrication Fischer–Burmeister–Newton–Schur (HMEHL-FBNS) solver implementation in MATLAB$^{©}$ was employed. It is an extension of the EHL-FBNS solver presented in an earlier work [23]. The homogenised Reynolds equation is discretised with the finite volume method (FVM), where the Poiseuille terms are discretised with a second-order central scheme and the Couette term with a first-order upwind scheme. The fundamental equations incorporated in the extended solver are summarised in the following.

The difference between the hydrodynamic pressure $p_{hd}$ and cavitation pressure $p_{cav}$ is called relative pressure $p = p_{hd} - p_{cav}$. The cavity fraction $\theta$ is defined as $\theta = 1 - \frac{\rho}{\rho_l}$, where $\rho$ is the mixture density of the flow and $\rho_l$ is the density of the liquid phase. $p$ and $\theta$ are determined at each position $(x_1, x_2)$ on the macroscopic scale with the steady homogenised Reynolds equation and the complementary cavitation constraint:

$$0 = \nabla \cdot \left( \frac{\rho_l h_m^3}{12 \mu_l} \boldsymbol{A} \, \nabla p - \rho_l h_m u_m \vec{b} \, (1 - \theta) \right), \tag{17}$$

$$p\theta = 0, \quad p \geq 0, \quad \theta \geq 0, \tag{18}$$

where the dynamic viscosity of the liquid phase $\mu_l$, mean velocity $u_m$, mean gap height $h_m$ and homogenisation factors $A$ and $\vec{b}$ are incorporated. For the pin-on-disk tribometer, the mean velocity is $u_m = U/2$, where $U$ is the velocity of the disk above the tribological contact. $h_m$ and the homogenisation factors $A$ and $\vec{b}$ are functions of the rigid body displacement on the roughness scale $h_0$. Their values and the mean contact pressure $p_{con}(h_0)$ can be interpolated at each position $(x_1, x_2)$ on the macroscopic scale by realising that the macroscopic gap height is the same as the rigid body displacement on the roughness scale $h_0$. If $h_0$ is larger than the $h_{0,r,max}$ during the interpolation process described in Section 2.2.1, then $h_m$ is set equal to $h_0$, the mean contact pressure is set to 0 and the homogenisation factors are set to their value at $h_{0,r,max}$, thus being close to either 0 or 1 depending on the factor. On the macroscopic scale, $h_0$ can be determined as:

$$h_0(x_1, x_2) = h_d + h_g(x_1, x_2) + h_{el}(x_1, x_2), \tag{19}$$

where $h_d$ denotes the rigid body displacement between the upper and lower macrogeometries, $h_g$ is the gap height variation due to the rigid macrogeometries and $h_{el}$ describes the combined elastic deformation of the macrogeometries. Using the elastic-half space assumption, $h_{el}$ is found to be:

$$h_{el}(x_1, x_2) = \frac{2}{\pi E'} \int\limits_{L_1} \int\limits_{L_2} \frac{p_{tot}(x'_1, x'_2)}{\sqrt{(x_1 - x'_1)^2 + (x_2 - x'_2)^2}} dx'_2 dx'_1, \tag{20}$$

where the total pressure

$$p_{tot} = p_{hd} + p_{con} \tag{21}$$

is a superposition of the hydrodynamic pressure $p_{hd}$ and the mean contact pressure $p_{con}$ within the domain of size $L_1 L_2$. Using Young's modulus $E$ and Poisson ratio $\nu$ of the upper and lower surfaces as described by the subscripts $_{up}$ and $_{low}$, respectively, the reduced elastic modulus is expressed as:

$$E' = \frac{2}{\frac{1-\nu_{low}^2}{E_{low}} + \frac{1-\nu_{up}^2}{E_{up}}}. \tag{22}$$

Considering the validity of the half-space assumption, Zhang et al. [24] investigated whether the free ends of the geometry should be considered or can be neglected in the modelling of the elastic deformation. They found that, for heavily loaded roller bearings under EHL conditions, neglecting the free ends delivers noticeable errors in the resulting deformation and pressure fields. However, these operating conditions differ fundamentally from those of the only lightly loaded conformal contact in the pin-on-disk tribometer. Even though there are free ends at rim of the pin, another work [7] indicated that they introduce only small deviations in the Stribeck curve for simulations with an imposed normal load and neglecting their effect on the elastic deformation is therefore legit.

Piezoviscosity of the liquid phase is incorporated with the Roelands equation:

$$\mu_l = \mu_0 \exp\left( (\ln(\mu_0) + 9.67) \cdot \left( -1 + \left( 1 + \frac{(p_{hd} - p_{cav})}{p_{0,R}} \right)^{z_R} \right) \right), \tag{23}$$

where $z_R = \frac{\alpha_R p_{0,R}}{\ln(\mu_0 + 9.67)}$ describes the pressure viscosity index, $\alpha_R$ is the pressure viscosity coefficient, $p_{0,R}$ denotes a constant in the Roelands equation and $\mu_0$ is the dynamic viscosity of the liquid phase at ambient pressure. The compressibility of the liquid phase is considered with the Dowson–Higginson model:

$$\rho_l = \rho_0 \frac{C_1 + C_2(p_{hd} - p_{cav})}{C_1 + (p_{hd} - p_{cav})}, \tag{24}$$

where $\rho_0$ is the density of the liquid phase at ambient pressure and $C_1$ and $C_2$ are constants. In summary, the consequences of the employed models for the dynamic viscosity and density are as follows: while each simulation for itself is isothermal, the lubricant's dependence upon temperature is still considered by the corresponding value of $\mu_0$ at 24 and 50 °C. The density is treated analogously. However, unlike $\mu_0$, the value of $\rho_0$ can actually be eliminated from the Reynolds Equation (17), because it appears in each single term. Consequently, the temperature dependence of $\rho_0$ does not need any special consideration in the employed model. Lastly, one estimate of the magnitude of $\rho_0$ is still necessary for the underlying EHL-FBNS solver described in [23] to create the non-dimensional equation system. Nonetheless, piezoviscous and compressible effects are considered by Equations (23) and (24). They allow for the dynamic viscosity and density to change throughout the simulation domain according to the hydrodynamic pressure field.

Then, how the mean contact pressure is added to the already published EHL-FBNS algorithm is described [23]. At each iteration $n$, the update of the mean contact pressure is computed as:

$$\delta_{p_{con}}^n = p_{con}(h_0) - p_{con}^{n-1},\tag{25}$$

where $p_{con}(h_0)$ is the expected mean contact pressure according to $h_0(x_1, x_2)$ and $p_{con}^{n-1}$ describes the mean contact pressure of the previous iteration. For stability reasons, this update is applied with an underrelaxation coefficient $\alpha_{p_{con}}$ to obtain the mean contact pressure at iteration $n$:

$$p_{con}^n = p_{con}^{n-1} + \alpha_{p_{con}}\delta_{p_{con}}^n \tag{26}$$

At the end of each iteration, $p_{con}^n$ is used to compute $p_{tot}$ according to Equation (21). Furthermore, the residual of $p_{con}$ is computed as:

$$r_{max,\delta p_{con}^*} = \max\left(\text{abs}\left(\frac{\delta_{p_{con}}^n}{p_{ref}}\right)\right).\tag{27}$$

This residual is added to [23] (Equation (28)) when the convergence of the whole algorithm is checked against a tolerance. Moreover, the load balance equation taking the ambient pressure $p_{amb}$ into account reads:

$$F_{N,imp} \overset{!}{=} F_N = \int_{L_1}\int_{L_2} p_{tot} - p_{amb}\,\mathrm{d}x_2\,\mathrm{d}x_1.\tag{28}$$

In the solver, the rigid body displacement between the upper and lower macroscopic geometries $h_d$ is adjusted through an incremental PID controller with its coefficients $K_P$, $K_I$ and $K_D$:

$$h_d^{n+1} = \Delta h_d^n + h_d^n = \left(K_P\left(r_{F_N}^n - r_{F_N}^{n-1}\right) + K_I r_{F_N}^n + K_D\left(r_{F_N}^n - 2r_{F_N}^{n-1} + r_{F_N}^{n-2}\right)\right)\cdot 10^{-6}\,\text{m} + h_d^n.\tag{29}$$

This was performed at each iteration until a prescribed tolerance was met by the residual between the resulting normal load $F_N$ and the imposed normal load $F_{N,imp} = 150$ N:

$$r_{F_N}^n = \frac{F_N^n - F_{N,imp}}{F_{N,imp}}.\tag{30}$$

Lastly, the hydrodynamic shear stresses on the pin surface are computed with the homogenised shear stress equation:

$$\vec{\tau}_{hd,3} = \begin{pmatrix}\tau_{hd,31}\\\tau_{hd,32}\end{pmatrix} = -\frac{h_m}{2}C\nabla p + \frac{\mu_l}{h_m}\left(-6u_m\vec{d} + u_r\begin{pmatrix}1\\0\end{pmatrix}\right)(1-\theta),\tag{31}$$

where $C$ and $\vec{d}$ are homogenisation factors and $u_r$ is the relative velocity between the upper and lower surfaces. For the pin-on-disk tribometer, $u_r = U$ holds. The mean contact shear stress is determined as:

$$\tau_{con} = C_{f,b} p_{con},\tag{32}$$

where $C_{f,b}$ is an estimate for the boundary friction coefficient. It is set to $C_{f,b} = 1/(3\sqrt{3})$, which is a value theoretically derived by Bowden and Tabor [25], Ch. V for the dry friction of metals under pure shearing, which serves as an unambiguous and uniquely defined upper limit for the boundary friction coefficient [7]. The total shear stress $\tau_{tot} = \tau_{hd,31} + \tau_{con}$ is computed by superposition and used to determine the resulting friction force $F_T$:

$$F_T = \int\limits_{L_1} \int\limits_{L_2} \tau_{tot}\, \mathrm{d}x_2\, \mathrm{d}x_1.\tag{33}$$

Eventually the friction coefficient is evaluated as:

$$C_f = \frac{F_T}{F_N}.\tag{34}$$

The parameter values used in the simulations are summarised in Table 6. The dynamic viscosity of the liquid phase $\mu_0$ at ambient pressure was set according to Table 3. The Dirichlet boundary conditions of ambient pressure $p_{amb}$ were employed for the hydrodynamic pressure $p_{hd}$ at the domain boundaries. Furthermore, the Dirichlet boundary condition of $\theta = 0$ was used at the domain inlet, whereas Neumann conditions were used for the cavity fraction at the remaining boundaries. A tolerance of $10^{-6}$ was used as a threshold for the residuals.

**Table 6.** Employed parameter values of the homogenised MEHL-FBNS solver. The parameters with indices $_{ref}$ are used to transform the steady homogenised Reynolds Equation (17) into a dimensionless form analogously to the procedure described in [23].

| Parameter | Value |
|:---:|:---:|
| $F_{N,imp}$ | 150 N |
| $U$ | $0.01\ldots 2$ m/s |
| $p_{amb}$ | $10^5$ Pa |
| $p_{cav}$ | $8 \cdot 10^4$ Pa |
| $\rho_0$ | $850$ kg/m$^3$ |
| $C_1$ | $5.9 \cdot 10^8$ Pa |
| $C_2$ | $1.34$ |
| $\alpha_R$ | $22 \cdot 10^{-9}$ /Pa |
| $p_{0,R}$ | $1.96 \cdot 10^8$ Pa |
| $\nu_{up}$ | $0.3$ |
| $\nu_{low}$ | $0.3$ |
| $E_{up}$ | $210 \cdot 10^9$ Pa |
| $E_{low}$ | $210 \cdot 10^9$ Pa |
| $\alpha_{p_{con}}$ | $0.05$ |
| $K_P$ | $1.2 \cdot 10^{-2}$ |
| $K_I$ | $2.4 \cdot 10^{-2}$ |
| $K_D$ | $1.5 \cdot 10^{-3}$ |
| $C_{f,b}$ | $1/(3\sqrt{3})$ |
| $x_{1,ref}$ | $8 \cdot 10^{-3}$ m |
| $x_{2,ref}$ | $8 \cdot 10^{-3}$ m |
| $h_{ref}$ | $10^{-6}$ m |
| $\mu_{ref}$ | $\mu_0$ |
| $\rho_{ref}$ | $\rho_0$ |
| $p_{ref}$ | $100 \cdot 10^6$ Pa |
| $u_{ref}$ | $u_m$ |

## 3. Results and Discussion

The general research strategy of this work was not to replicate each single experiment with a simulation, but instead to create a single virtual reference geometry which is able to replicate the friction behaviour of numerous experiments, thus being a representative digital twin of an arbitrary pin-on-disk specimen combination in the tribometer. The definition process of this representative geometry is referred to as calibration. In this section, the experiment test results are firstly used to define the set averages along with their deviations as an estimate of the scatter. Then, the calibration of the digital twin is performed and deviations of the Stribeck curve due to geometry variations are quantified to identify sensitivities. Subsequently, the Hersey number is derived for the considered pin-on-disk tribometer to allow a meaningful comparison of the experiment and simulation for the validation of the presented digital twin.

### 3.1. Experiment Results

In order to quantify the deviation of the Stribeck curves from a reference Stribeck curve $C_{f,ref}$ in relation to the overall magnitude of the boundary friction coefficient $C_{f,b}$, the deviation of the friction coefficient is defined as:

$$C_{f,dev} = \frac{C_f - C_{f,ref}}{C_{f,b}}. \tag{35}$$

The Stribeck curves of the three tests in set 1 at 24 °C along with their set average are depicted in Figure 7a. The area enclosed by the maximum and minimum friction coefficients at each relative velocity is shaded in grey to visualise the set scatter. The corresponding deviation of the friction coefficient is shown in Figure 7b, where the averaged Stribeck curve of the set is used as reference $C_{f,ref}$. The analogous results of set 2 at 50 °C are represented in Figure 8a,b and demonstrate that the largest deviation in the friction coefficient in the experiments is approximately $C_{f,dev,exp} \approx \pm 10\%$ and occurs in the mixed lubrication regime.

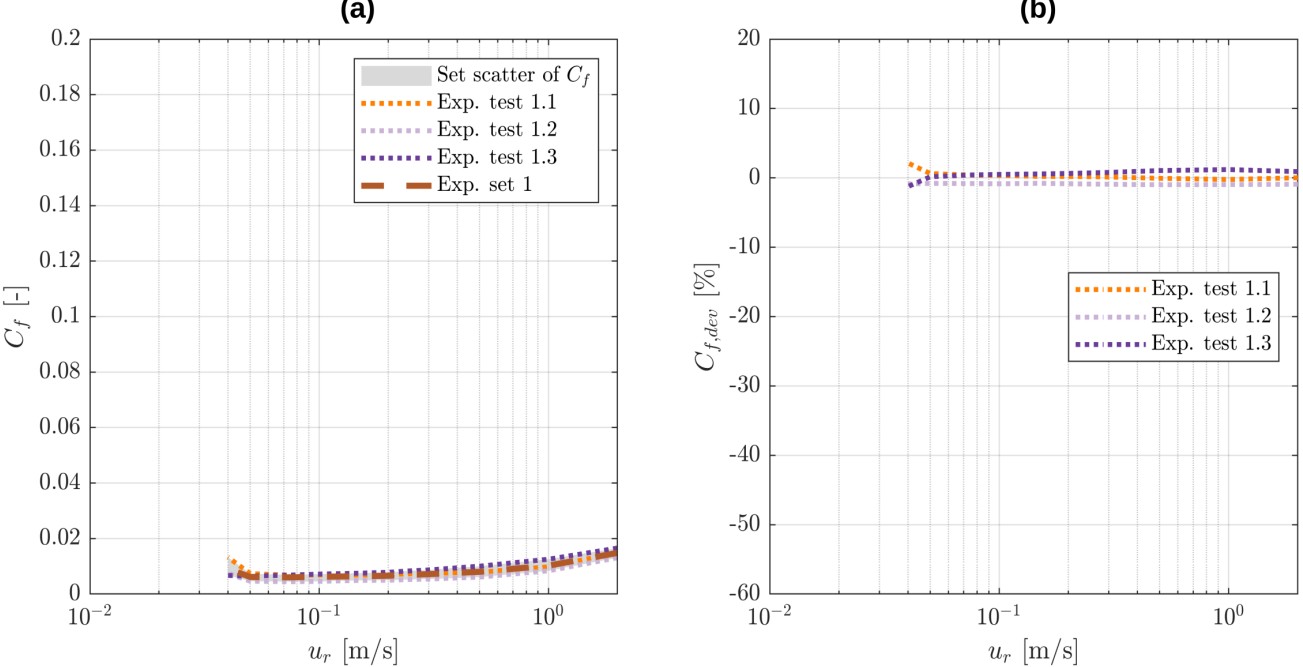

**Figure 7.** Experimental results of the tests in set 1 and their set average. (**a**) Friction coefficient $C_f$ as a function of relative velocity $u_r$. (**b**) Deviation of the friction coefficient $C_{f,dev}$ as a function of relative velocity $u_r$.

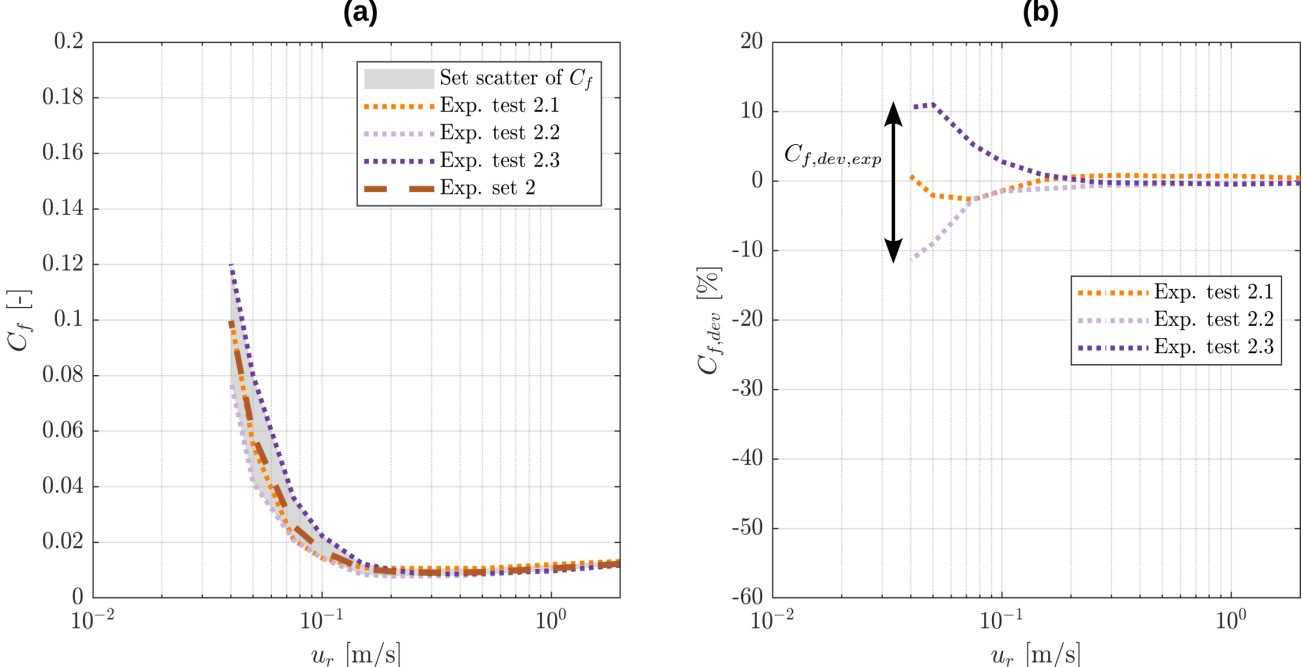

**Figure 8.** Experimental results of the tests in set 2 and their set average. (**a**) Friction coefficient $C_f$ as a function of relative velocity $u_r$. (**b**) Deviation of the friction coefficient $C_{f,dev}$ as a function of relative velocity $u_r$.

### 3.2. Calibration

Firstly, the size of the Gaussian filter for the smoothing of the macroscopic topographies is evaluated. The filter size should be large enough to smooth out all roughness effects but small enough to prevent the filtering of macroscopic geometry features. The size of the Gaussian filter is parametrised by its full width at half maximum of the Gaussian distribution. The used macroscopic pin and disk topographies have a resolution of $500 \times 500$ points. The roughness properties of patch 4 of test 2.1 are used in this case. The simulated Stribeck curves for the unfiltered profile and filter sizes of 3, 5, 7, 9 and 11 pixel are displayed in Figure 9a. This demonstrates that the smoothing of the geometry on the macroscopic scale causes the Stribeck curve to be shifted to lower relative velocities $u_r$.

In order to quantify the resulting deviations in the Stribeck curve, the 11-pixel filter size simulation is chosen as the reference $C_{f,ref}$. The results of $C_{f,dev}$ are displayed in Figure 9b. The deviations are the largest in the mixed lubrication regime. It is shown that the unfiltered profile can cause a deviation in the Stribeck curve of 45%. This is because the unfiltered macroscopic profile still contains roughness information which is already considered by the microscopic roughness scale. This wrongly creates a double implementation of roughness and causes a large shift in the Stribeck curve to higher relative velocities $u_r$. With increasing filter sizes, the Stribeck curves converge towards the reference solution. For a filter size of 9 pixels, the deviation becomes less than 0.5% which is chosen as the filter size for the following simulations.

The calibration of the macroscopic resolution is performed next by interpolating new geometries with the resolutions $128 \times 128$, $256 \times 256$, $999 \times 999$ and $1997 \times 1997$ from the one with a resolution of $500 \times 500$. The higher resolutions of $999 \times 999$ and $1997 \times 1997$ are chosen such that they exhibit all of the original $500 \times 500$ points with additional points in between. In order to save computational resources, the simulations for those resolutions were only conducted for relative velocities of 0.02, 0.04, 0.06, 0.08 and 0.1 m/s. The resulting Stribeck curves are shown in Figure 10a while the deviations of the friction coefficient $C_{f,dev}$ are depicted in Figure 10b. The Stribeck curve with the original resolution of $500 \times 500$ is chosen as $C_{f,ref}$. For the highest resolution, the maximum deviation is at 1.2%, while for a resolution of $256 \times 256$, its maximum absolute value is less than 1.8%.

Comparing the resolution of $256 \times 256$ to $1997 \times 1997$ thus yields an estimate of a deviation of less than 3%, which is considerably less than the $\pm 10\%$ of the experiments. Therefore, a resolution of $256 \times 256$ is considered sufficient and is used for the following simulations.

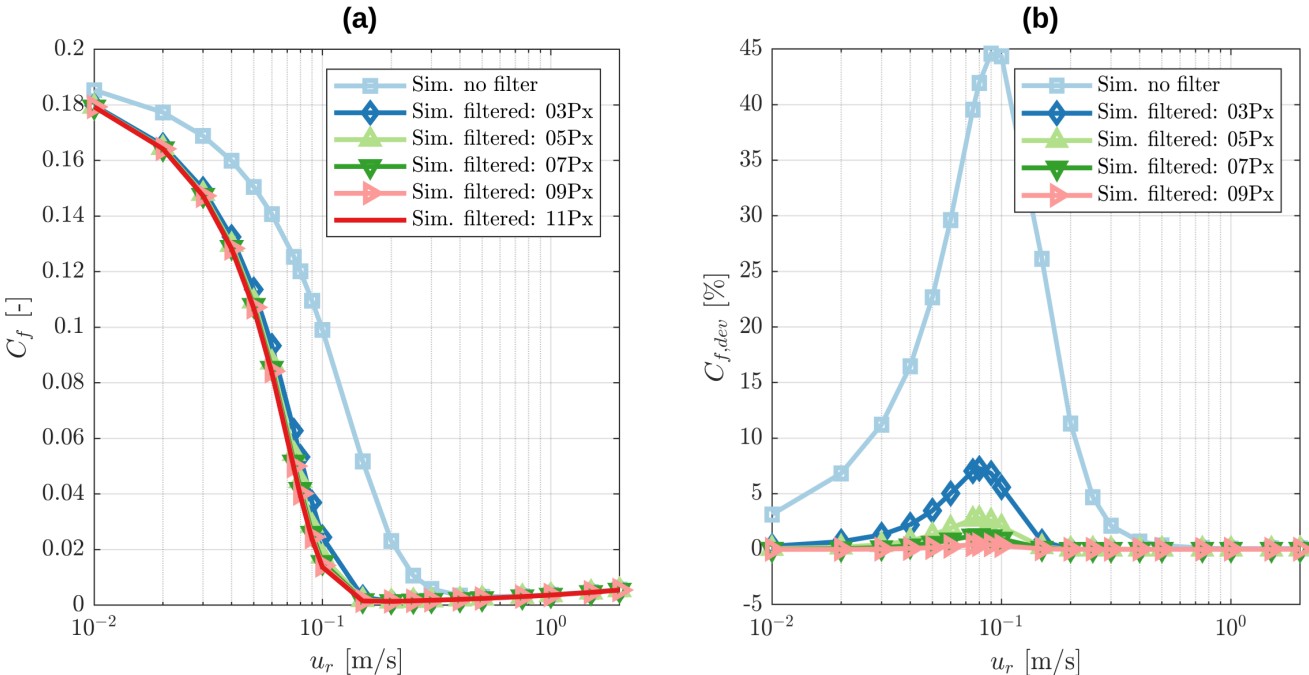

**Figure 9.** Simulation results for different filter sizes: (**a**) Friction coefficient $C_f$ as a function of relative velocity $u_r$; and (**b**) Deviation of the friction coefficient $C_{f,dev}$ as a function of relative velocity $u_r$.

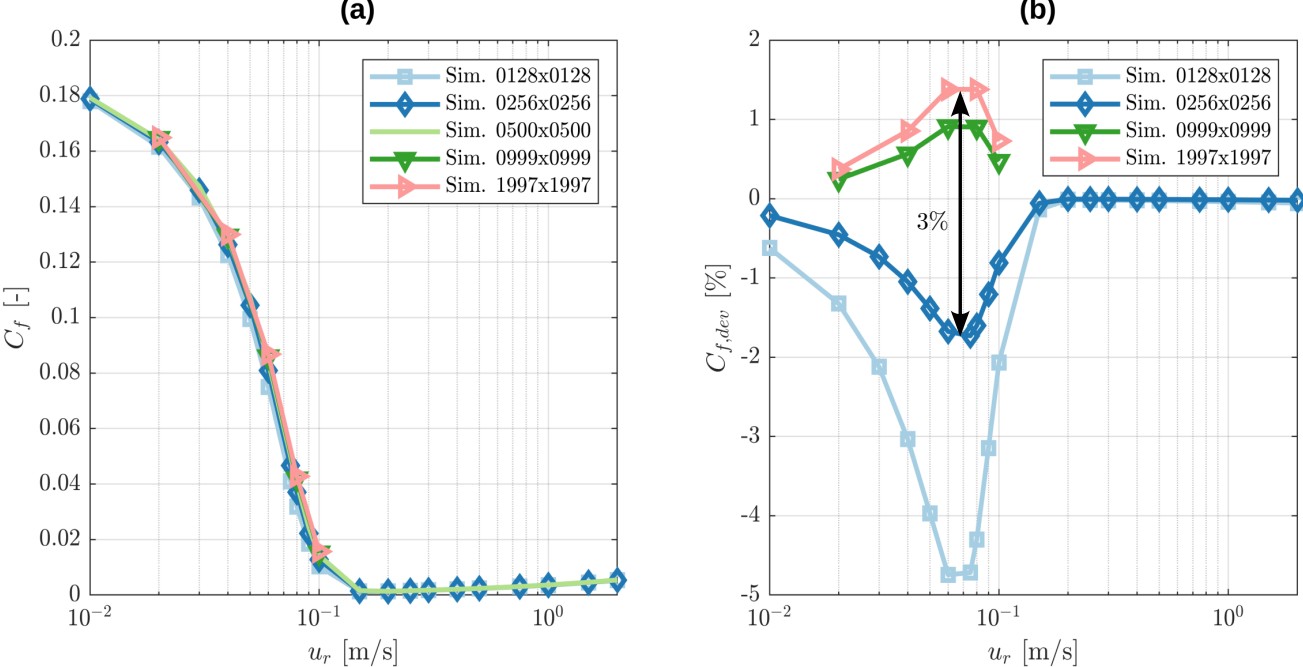

**Figure 10.** Simulation results for different resolutions. (**a**) Friction coefficient $C_f$ as a function of relative velocity $u_r$. (**b**) Deviation of the friction coefficient $C_{f,dev}$ as a function of relative velocity $u_r$.

Moreover, it is evaluated whether the macroscopic wear track on the disc influences the friction behaviour. Figure 11a shows the simulation results obtained when both the macroscopic pin and disk topographies are used to reconstruct the virtual geometry against

the idealised case where the disk is assumed to be perfectly flat on the macroscopic scale. Figure 11b shows that this only causes a maximum absolute deviation in the friction coefficient of less than 4%, where the case with non-flat disk is chosen as $C_{f,ref}$. Assuming that the disk is flat is therefore legit in comparison to the deviation in the experiments and performed for the following simulations. Furthermore, being able to neglect the macroscopic disc topography bears the significant advantage of a generally much simpler virtual geometry reconstruction workflow.

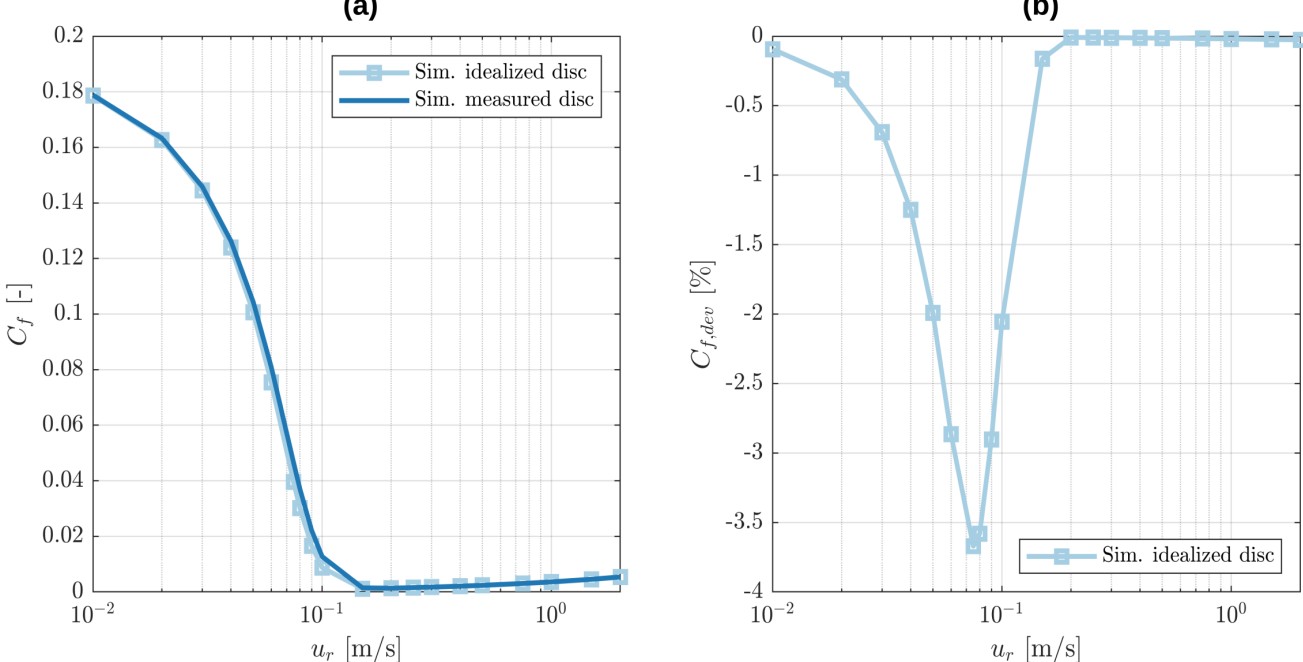

**Figure 11.** Simulation results with idealised and measured macroscopic disk topography. (**a**) Friction coefficient $C_f$ as a function of relative velocity $u_r$. (**b**) Deviation of the friction coefficient $C_{f,dev}$ as a function of relative velocity $u_r$.

The last step of calibrating the macroscopic geometry consists of averaging the macroscopic pin geometries of set 2, as shown previously in Figure 3. The Stribeck curves for the different topographies of tests 2.1, 2.2 and 2.3 and the average pin geometry of set 2 are depicted in Figure 12a. When the average pin geometry of set 2 is used as $C_{f,ref}$, the absolute deviations in the friction coefficient shown in Figure 12b are less than 4%. This low value firstly allows the very important deduction that the digital twin is robust with respect to the macroscopic measurement of the pin and that the macroscopic geometry obtained by averaging over the pins of set 2 is suitable for the following simulations. Secondly, the comparison to the deviations of $\pm 10\%$ in the experiments suggests that the deviations in the real-life measurements are unlikely to be caused by variations in the macroscopic geometry.

Finally, the microscopic calibration of the digital twin is performed. The Stribeck curves are simulated for the six different roughness patches obtained from test 2.1 and for the average of their roughness factors. The used roughness factors were shown earlier in Figures 4 and 5. The results are shown in Figure 13a. The deviations in the Stribeck curves are shown in Figure 13b, where the Stribeck curve based on the averaged roughness factors is used as $C_{f,ref}$. It becomes obvious that the digital twin is highly sensitive to the chosen roughness patch. While some patches deliver results closely the reference, other patches show deviations of up to 52%. In the mixed lubrication regime, the deviation of most roughness patches is of similar magnitude as the deviations within the experiment tests. This in turn suggests that the deviations in the experiments might be caused by differences in the roughnesses between the tests. At the same time, the variations in the roughnesses within test 2.1 also indicate that an actual tracing of exactly which roughness

patch at which position on the pin surface dominates the behaviour of the Stribeck curve is extremely difficult and beyond the scope of this work. Instead, in the following, this work aims to investigate how suitable the average of the roughness factors of test 2.1 is for representing the microscopic geometry of the digital twin.

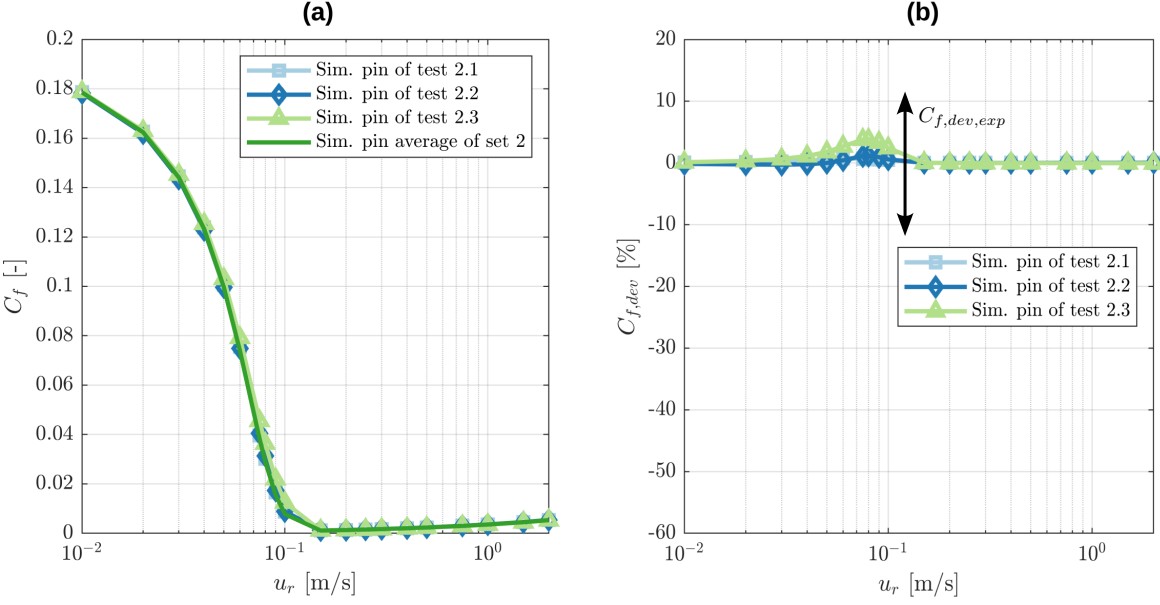

**Figure 12.** Simulation results for different measured macroscopic pin topographies and their set average. (**a**) Friction coefficient $C_f$ as a function of relative velocity $u_r$. (**b**) Deviation of the friction coefficient $C_{f,dev}$ as a function of relative velocity $u_r$.

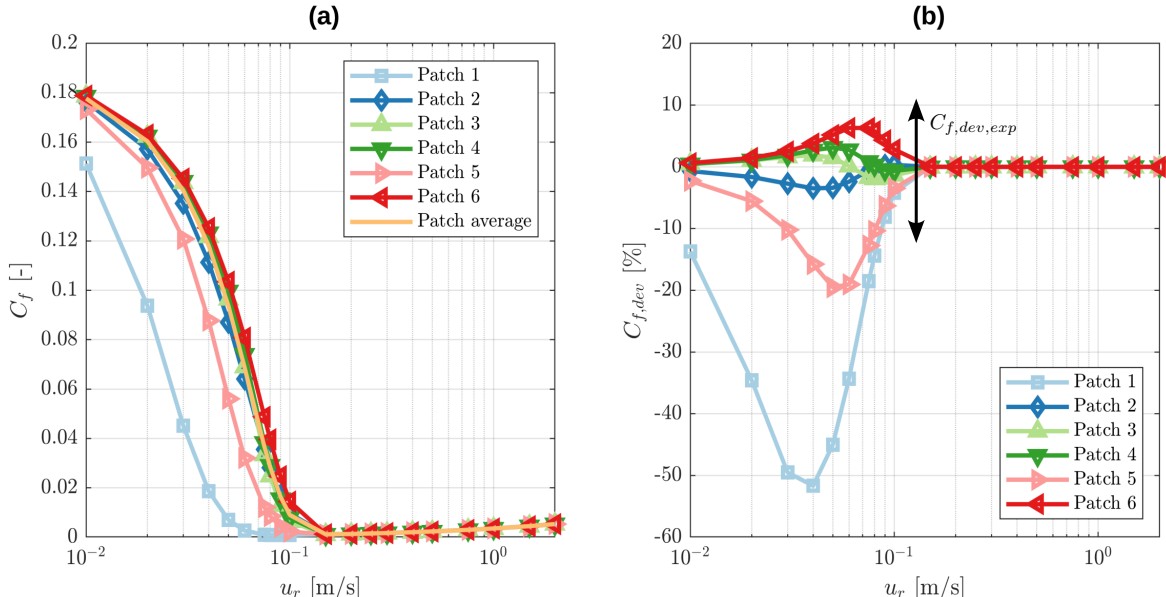

**Figure 13.** Simulation results for different measured microscopic patches and their average. (**a**) Friction coefficient $C_f$ as a function of relative velocity $u_r$. (**b**) Deviation of the friction coefficient $C_{f,dev}$ as a function of relative velocity $u_r$.

### 3.3. Validation

As a useful tool for the validation of the calibrated virtual geometry, the Hersey number [26] (Ch. 1.3.6), [27] for the considered pin-on-disk tribometer is derived to allow

a certain degree of generalisation of the obtained results. Starting point is the steady two-dimensional Reynolds equation, for simplicity without homogenisation factors:

$$\nabla \cdot \left(\frac{\rho_l h^3}{12\mu_l}\nabla p\right) - \frac{\partial}{\partial x_1}(\rho_l h u_m (1-\theta))) = 0. \tag{36}$$

An order of magnitude analysis can be performed when each dimensional variable and parameter is decomposed into its dimensionless value denoted by * and its reference value denoted by $_{ref}$:

$$h = h^* h_{ref}; \quad p = p^* p_{ref}; \quad \rho_l = \rho^* \rho_{ref}; \quad \mu_l = \mu^* \mu_{ref}; \quad x_1 = x_1^* x_{1,ref}; \quad x_2 = x_2^* x_{2,ref}; \quad u_m = u_m^* u_{m,ref}. \tag{37}$$

where the magnitudes of the dimensional values are roughly estimated as follows:

$$\begin{aligned} p_{ref} = F_{N,imp}/D^2; \quad \rho_{ref} = \rho_0; \quad \mu_{ref} = \mu_0; \\ x_{1,ref} = D; \quad x_{2,ref} = D; \quad u_{m,ref} = U/2, \end{aligned} \tag{38}$$

where $D$ is the diameter of the pin. Since $h$ is adjusted by $h_d$ in order to fulfil the load balance equation, its magnitude $h_{ref}$ is unknown and cannot be directly estimated. To rectify this, the Reynolds equation is put into non-dimensional form using both the estimated and the unknown magnitudes:

$$\nabla^* \cdot \left(\frac{\rho^* h^{*3}}{\mu^*}\nabla p^*\right) - 6\mathcal{S}\frac{\partial}{\partial x_1^*}(\rho^* h^* u_m^* (1-\theta)) = 0, \tag{39}$$

which firstly allows to define the Sommerfeld number [26] (Ch. 11.2) [28] of the pin-on-disk tribometer:

$$\mathcal{S} = \frac{\mu_0 U D^3}{h_{ref}^2 F_{N,imp}}. \tag{40}$$

The idea of the non-dimensionalisation is that all dimensionless variables and parameters are approximately of magnitude 1 if the magnitudes of their dimensional counterparts are properly estimated. This, in turn, allows to approximately deduce the magnitude of the gap height $h$ from the dimensionless Reynolds Equation (39) as:

$$h_{ref} \approx \sqrt{\frac{\mu_0 U D^3}{F_{N,imp}}}. \tag{41}$$

Next, the steady hydrodynamic shear stress equation is considered:

$$\tau_{hd,31} = -\frac{h}{2}\frac{\partial p}{\partial x_1} + \frac{\mu_l u_r}{h}(1-\theta). \tag{42}$$

Using $u_r = u_r^* u_{r,ref}$ with $u_{r,ref} = U$ and reformulating the right side of the above equation yields:

$$\tau_{hd,31} = \sqrt{\frac{\mu_0 U F_{N,imp}}{D^3}}\left(-\frac{h^*}{2}\frac{\partial p^*}{\partial x_1^*} + \frac{\mu^* u_r^*}{h^*}(1-\theta)\right), \tag{43}$$

which allows one to deduce the magnitude of the hydrodynamic shear stress as:

$$\tau_{hd,31,ref} = \sqrt{\frac{\mu_0 U F_{N,imp}}{D^3}}. \tag{44}$$

Lastly, the scaling behaviour of the hydrodynamic friction coefficient can be found by estimating its magnitude using $F_{T,hd,ref} = \tau_{hd,31,ref}D^2$:

$$C_{f,hd,ref} = \frac{F_{T,hd,ref}}{F_{N,imp}} = \sqrt{\mathcal{H}}, \tag{45}$$

where

$$\mathcal{H} = \frac{\mu_0 U D}{F_{N,imp}}, \tag{46}$$

is the Hersey number [26] (Ch. 1.3.6) of the pin-on-disk tribometer. An equivalent expression of $\mathcal{H}$ for the horizontal journal bearings under steady and fully lubricated conditions was first derived by Hersey [27] in 1914 using the Buckingham $\Pi$ theorem. The derived Hersey number in Equation (46) provides a basis for a very particular theoretical understanding of the hydrodynamics of the lubricant flow: the imposed normal load and thus the hydrodynamic load carrying capacity are directly proportional to the dynamic viscosity of the lubricant.

For validation, the defined reference geometry obtained with a nine-pixel filter, $256 \times 256$ resolution, smooth macroscopic disk, macroscopic pin average of set 2 and roughness factor average over patches 1–6 of test 2.1 was employed for simulations with the oil's respective dynamic viscosities at 24 and 50 °C. The simulation results are plotted in comparison to the experiment Stribeck curves of sets 1 and 2 in Figure 14a as a function of the relative velocity $u_r$ and in Figure 14b as a function of the Hersey number $\mathcal{H}$. When the Hersey number is used, all results collapse on one curve, which firstly shows that the dynamic viscosity proportionality is applicable to both the simulation and the experiment. Secondly, this demonstrates that the model is capable of predicting the transition from the hydrodynamic to the mixed lubrication regime. While the curve collapsing of the simulation results is a consequence of the underlying equations and thus only a verification of a correct implementation, the agreement of the collapsed simulation with the experiment Stribeck curves validates the digital twin as a tool to predict the friction behaviour of a pin-on-disk tribometer in the mixed lubrication regime.

Furthermore, the following line of argumentation can be drawn for the mixed lubrication regime. According to the employed model, the friction coefficient in the mixed lubrication regime is dominantly determined by the surface contact pressure, which in turn is determined by the load carrying capacity of the lubricant. The reason for this is that the lubricant pressure build up separates the surfaces as much as it is can while the remaining load is carried by the contact pressure which then causes the large contact shear stress.

In essence, this means that the friction coefficient in the mixed lubrication regime is strongly influenced by the load carrying capacity of the lubricant flow. Keeping in mind that the mixed lubrication regime can consistently be represented by a collapsed curve for the experiments and the simulations, this leads to the confirmation that the hydrodynamic load carrying capacity is indeed proportional to the dynamic viscosity of the lubricant, as predicted earlier by Equation (46). Finally, this allows us to conclude that the presented digital twin can properly predict the load carrying capacity of the real-life lubricant flow and is a valid tool to gain valuable insights into tribological systems.

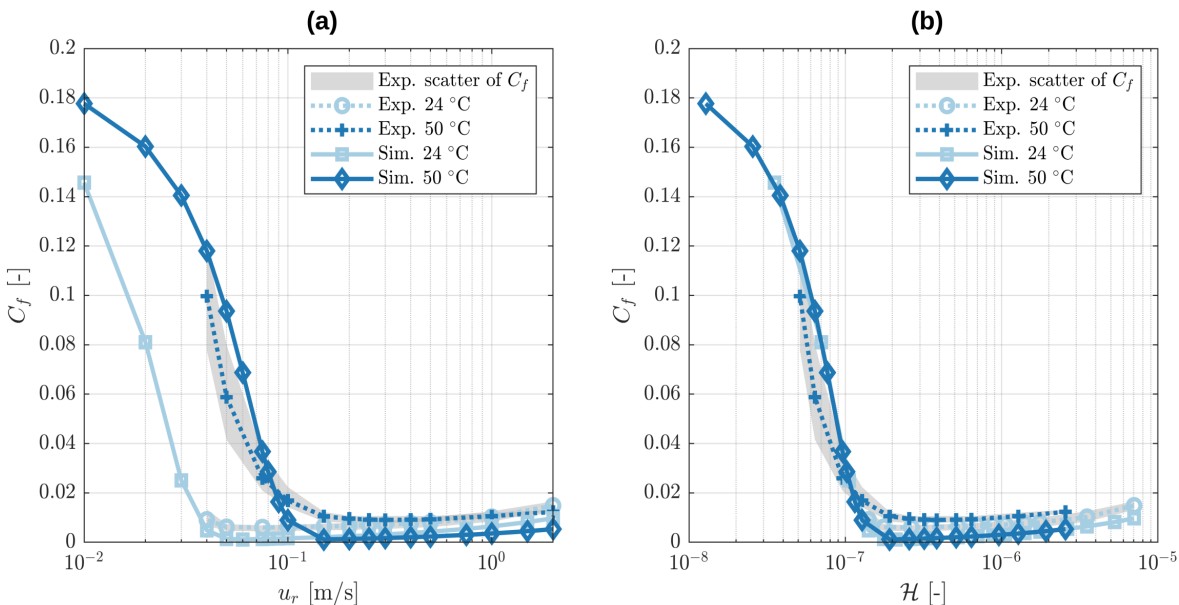

**Figure 14.** Stribeck curves for different temperatures. Full lines: simulation results. Dashed lines: experiment results of set averages. (**a**) Friction coefficient $C_f$ as a function of relative velocity $u_r$. (**b**) Friction coefficient $C_f$ as a function of Hersey number $\mathcal{H}$.

## 4. Conclusions

In this investigation, a procedure for the establishment and subsequent calibration of a digital twin of a pin-on-disk tribometer was presented. Particularly noteworthy is the fact that its virtual geometry is not approximated by parametrised shapes, but instead, it is completely deduced from real-life topography measurements. The advantage of this approach was that the described procedure can be applied to any geometry, even one that is too complex to be parametrised, as long as its topographies can be measured. The presented work includes a description of the real-life experimental procedure, the topography measurements of the specimen, the reconstruction of the virtual geometry and the employed multi-scale mixed lubrication solver along with exemplary data and code in the supplements. Furthermore, the Hersey number was derived to generalise findings about the hydrodynamic load carrying capacity and friction behaviour of the pin-on-disk tribometer and to allow the verification and validation of the digital twin with the experimental results. During the conducted evaluation of the experiment and simulation of the Stribeck curves, the following key statements were drawn:

- Filtering of the macroscopic topographies is an essential step for multi-scale solvers to prevent a double consideration of roughness. After filtering, the digital twin is very robust with regard to variations in the macroscopic geometry of different specimens. The macroscopic geometry of the disc is even negligible.
- The digital twin is highly sensitive to the employed roughness patch. Averaging over the computed influence factors of several roughness patches was shown to deliver good agreement with the experiment results. The deviations of simulations for different roughness patches are of similar magnitude as the deviations of the experiments within one set, thus suggesting that the deviations in the experiment are caused by variations in the roughness.
- The hydrodynamic load carrying capacity scales proportionally with the dynamic viscosity of the lubricant. This leads to a collapse of Stribeck curves in the mixed lubrication regime when the friction coefficient is plotted as a function of the Hersey number.

**Supplementary Materials:** The following supporting information can be downloaded at: https: //doi.org/10.5281/zenodo.7540491.

**Author Contributions:** Conceptualisation, E.H., G.V., J.S., P.G. and B.F.; methodology, E.H. and G.V.; software, E.H.; validation, E.H. and G.V.; formal analysis, E.H.; investigation, E.H. and G.V.; resources, B.F. and P.G.; data curation, E.H. and G.V.; writing—original draft preparation, E.H. and G.V.; writing—review and editing, E.H., G.V., J.S., P.G. and B.F.; visualisation, E.H.; supervision, J.S., P.G. and B.F.; project administration, E.H., G.V., J.S., P.G. and B.F.; funding acquisition, P.G. and B.F. All authors have read and agreed to the published version of the manuscript.

**Funding:** This research was funded by Deutsche Forschungsgemeinschaft (DFG) Project Number 438122912. We acknowledge support by the KIT-Publication Fund of the Karlsruhe Institute of Technology.

**Data Availability Statement:** Code examples for the macroscopic averaging including the subsequent pin levelling, roughness influence factor computation and the homogenised MEHL-FBNS solver are supplied on https://doi.org/10.5281/zenodo.7540491 (accessed on 16 January 2023) along with some of the virtual geometries and influence factors. The codes are ready to execute exemplary computations. The used contact solvers are based on implementations which are publicly available on GitHub: https://github.com/ErikHansenGit/Contact_elastic_half-space (accessed on 16 January 2023).

**Conflicts of Interest:** The funders had no role in the design of the study; in the collection, analyses, or interpretation of data; in the writing of the manuscript; or in the decision to publish the results.

## Abbreviations

The following abbreviations are used in this manuscript:

| | |
|---|---|
| CG-FFT | Conjugate Gradient-Fast Fourier Transform |
| EHL | Elasto-Hydrodynamic Lubrication |
| EHL-FBNS | Elasto-Hydrodynamic Lubrication-Fischer–Burmeister–Newton–Schur |
| HMEHL-FBNS | Homogenised Mixed Elasto-Hydrodynamic Lubrication-Fischer–Burmeister-Newton–Schur |

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
