# Peer review of "Establishment and Calibration of a Digital Twin to Replicate the Friction Behaviour of a Pin-on-Disk Tribometer"

_lubricants, doi:10.3390/lubricants11020075_

Round 1

Reviewer 1 Report

This paper has proposed an establishment of a digital twin of a pin-on-disc tribometer whose virtual geometry is replicated from real-life post-test topography measurements and fed into a two-scale mixed lubrication solver. The paper was well prepared, with a comprehensive result analysis and discussion. However, there is an obvious disadvantage that only one load of 50 N and two temperatures (24 and 50 oC) have been considered. The authors should provide more relevant studies on these two factors which are very important in practical friction contacts. After addressing these two issues, the paper can then be accepted for publication.

Author Response

Dear reviewer,

thank you for your interesting perspective and valuable comment. The employed operating conditions are chosen such that the occurring wear does not exceed simple running in effects. The reason is that we aim at having a geometry as similar as possible under all operating conditions to isolate the effect due to changes in the dynamic viscosity. At temperatures and loads above the employed ones, wear debris starts to form and large wear scars appear. These notably alter the macroscopic geometry and render a comparison to the already obtained results difficult because they cannot be pinned down anymore only to the viscosity changes. Additionally, the wear effects would require a different solver configuration. This is the reason why we prefer to stay within in the already employed operating conditions in the scope of this work and leave the suggested extension to future investigations.

A comment on this was added in the manuscript in lines 106-109. 

Kind regards
the authors

Reviewer 2 Report

This is an interesting study concerning the possibility of designing digital twins of a wear testing syste. Excellent agreement between the simulation and the experimental results was found. However, the application of th paper would be improved if the authors could discuss the applicability of their approach to wear situations where wear to either of the contacting surfaces occurs. It would also be interesting to see the authors views on the application of the approach to specific lubricated applications rather than model testing systems.

Author Response

Dear reviewer,

thank you for your additional perspective for wear testing systems. However, the employed operating conditions are intentionally chosen such that the occurring wear does not exceed simple running in effects. We agree that an applicability to wear situations would be of great interest, however this would also require both an adjustment of the experiment procedure and the numerical model to properly evaluate the additional wear effects. This in turn would be out of the scope of this work. In order to stay where the conducted work can deliver reliable results, we would therefore prefer not to suggest any extrapolation of our results to wear conditions.
A comment on this was added in the manuscript in lines 106-109. 

Our perspective is similar on the extrapolation to specific lubricated applications. It seems likely that any application which exhibits the same operating conditions as a pin-on-disk tribometer could be investigated with a digital twin that is analogous to ours. Moreover, an advantage of our procedure is that the it can be applied to any geometry as long as its topographies can be measured. However, the authors do not dare yet to suggest it as a general procedure without any adaptations for specific applications because this would priorly require a separate analysis of the respective application.
Remarks on this were added in the manuscript in lines 3-6 and 487-493.

Kind regards
the authors

Reviewer 3 Report

Recommendation: Minor Revision

Below is a list of detailed comments and questions.

1.     The originality of the paper needs to be further clarified in the Abstract. Please, involve the novelty of this paper not what you have done in this study.

2.     In the simulation section, Please, mention all assumptions for building the current model to present the most relevant considerations studied.

3.      The effect of temperature on lube oil viscosity requires further explanation. Also, the effect of temperature and pressure on oil density needs to be detailed. Two influential properties in the Reynolds equation are the density and viscosity of the lube oil.

4.     In Conclusion, what is the critical contribution of this study to knowledge in the tribology field?

Author Response

Dear reviewer,

thank you for your valuable suggestions and helpful comments. The originality of our work lies in the complete deduction of the virtual geometry from real-life topographies, both on the micro- and the macro-scale. In contrast, most of the approaches in the literature often approximate the virtual geometries with simplified parameterized analytical functions. The advantage of the presented approach is that the described procedure can be applied to any geometry, even one that its too complex to be parameterized, as long as its topographies can be measured.
Remarks on this were added in the manuscript in lines 3-6 and 487-493.

Furthermore, the effect of temperature and pressure on viscosity and density due to the assumptions that come with the employed model were further elaborated in lines 285-295.

Kind regards
the authors

Round 2

Reviewer 1 Report

The revised paper can be accepted for publication in the present form.